
Materials Science

# Limited proteolysis of pyranose 2-oxidase results in a stable and active complex

Tanzila Islam, Catherine Booker, Dmitri Tolkatchev, Su Ha and Alla S. Kostyukova

Voiland School of Chemical Engineering and Bioengineering, Washington State University, Pullman, WA, United States of America

## ABSTRACT

Glucose oxidating enzymes have a tremendous potential for various energy, healthcare and environmental sensing applications. In this work, we studied the effect of reducing the size of pyranose 2-oxidase (POx) on stability and enzymatic activity of proteolyzed POx. Limited proteolysis of the POx was performed using trypsin to remove flexible structural regions without significant damage to the overall conformation and catalytic activity of the enzyme. Enzymatic activities of the modified and wild-type POx were measured by colorimetric coupled reaction assay and compared. The enzymatic activity of the modified POx showed 90% activity compared to the wild-type POx. This result indicates that reducing the size of the protein can be done without losing enzymatic activity and such enzymes potentially could provide a larger gain in electrochemical activity compared with wild-type enzymes.

## INTRODUCTION

The combination of proteins and nanomaterials to produce various functional enzyme electrodes is important for both understanding the fundamental protein-nanomaterial interactions and the development of high performance bioelectronic devices (*Willner & Katz, 2005*). An establishment of an efficient electrochemical communication between the active site of protein and the electrode surfaces, while maintaining a long-term protein stability, is a critical step for developments of any practical bioelectronics devices (e.g., biosensors (*Zhu et al., 2015*), biofuel cells (*Atanassov et al., 2007*), neuron-semiconductor hybrid systems for dynamic memory (*Willner & Katz, 2005*), etc.). To achieve these goals, previous investigators have created functional enzyme electrodes by immobilizing the enzyme on the external surface of nanomaterials (*Fischback et al., 2006*; *Ivnitski, Atanassov & Apblett, 2007*; *Kim, Jia & Wang, 2006*; *Jia et al., 2002*; *Campbell et al., 2015*; *Scherbahn et al., 2014*; *Prasad, Chen & Chen, 2014*); by entrapping it in sol-gel (*Kim, Jia & Wang, 2006*; *Galarneau et al., 2007*; *Nogala et al., 2008*; *Lim et al., 2007*) or polymers (*Atanassov et al., 2007*; *Moore et al., 2004*; *Mano, Mao & Heller, 2003*; *Mano et al., 2003*; *Mao, Mano & Heller, 2003*; *Akers, Moore & Minteer, 2005*; *Xu et al., 2014*; *Aquino Neto et al., 2015*; *Aquino Neto et al., 2014*; *Gonzalez-Guerrero et al., 2013*), by using bulk composite (*Kim, Jia & Wang, 2006*); by chemically modifying structures of enzymes (*Willner, Katz &*

Corresponding authors
Su Ha, su.ha@wsu.edu
Alla S. Kostyukova,
alla.kostyukova@wsu.edu

*Willner, 2006*; *Zayats et al., 2005*; *Katz, Sheeney-Haj-Ichia & Willner, 2004*; *Katz & Willner, 2003*; *Katz & MacVittie, 2013*). However, either a poor electrochemical communication of these immobilized enzymes to electrodes or their short lifetime hinders the development of practical bioelectronic devices (*Atanassov et al., 2007*; *Kang et al., 2004*; *Ha, Wee & Kim, 2012*).

Effective direct, non-mediated electrical wiring of enzyme on electrodes remains a challenging task (*Freire et al., 2003*; *Marcus & Sutin, 1985*). Newly developed approaches that can enhance direct electron transfer include improving the surface properties of the electrode by introducing highly conductive nano-particles and controlling enzyme orientation by changing surface charge properties (*Lalaoui et al., 2016a*; *Lalaoui et al., 2016b*; *Gross et al., 2017a*). Several recent studies developed modified enzyme supports for better wiring by surface grafting of nanoparticle supports with mediators (*Moore et al., 2004*; *Lalaoui et al., 2016a*; *Gross et al., 2017a*; *Cosnier et al., 2016*; *Bourourou et al., 2014*; *Gross et al., 2017b*; *Rengaraj et al., 2017*; *Isikli et al., 2014*; *Klotzbach et al., 2006*; *Monsalve et al., 2015*; *Mazurenko et al., 2016*). Some research groups were able to modify the surface charge properties of enzymes to control the orientation of their adsorption on electrode surface (*Lalaoui et al., 2016a*; *Lalaoui et al., 2016b*).

Glucose oxidase (GOx) is one of the most frequently used enzymes for glucose-based biofuel cells and biosensors due to its high and selective reactivity towards glucose oxidation, well-known crystal structure and easy availability. Recently, another flavin-dependent enzyme, pyranose 2-oxidase (POx), has gained increased attention due to its attractive structural features (*Kwon et al., 2014*; *Tasca et al., 2010*). POx catalyzes the oxidation of D-glucose at C2 position to yield 2-dehydro D-glucose while producing electrons that can be utilized in either power generation for biofuel cells or glucose detection for the biosensors (*Schafer et al., 1996*; *Wongnate & Chaiyen, 2013*; *Tan et al., 2014*). Unlike GOx, the POx protein is not surrounded by electrochemically nonconductive glycosylation layer in its native state. This structural characteristic offers better mass transport of glucose and better potential for efficient electron transfer from the active site to the electrode. Therefore, POx is considered a good candidate for biofuel cell and biosensor applications.

POx is a homotetrameric enzyme, which contains one flavin adenine dinucleotide (FAD) per subunit (*Hassan et al., 2013*; *Danneel et al., 1993*). POx tetramer structure has a hydrodynamic radius of 6.2 nm where each of the 4 FAD/FADH$_2$ centers is buried ∼1.4 nm below the enzyme surface (*Danneel et al., 1993*; *Martin Hallberg et al., 2004*). In our current study, we aimed to remove its exposed flexible polypeptide regions. The main idea was to decrease the POx dimensions in such a way that the protein core remains intact. We chose POx from *Phanerochaete chrysosporium* because it is the most stable POx compared to POx from other organisms (*Salaheddin et al., 2010*). The exposed flexible regions were removed from the surface of POx by limited proteolysis. We demonstrated that the POx structure minimization by limited trypsinolysis does not eliminate its enzymatic activity and the tryptic fragments still maintain a stable quaternary structure.

## MATERIALS & METHODS

### POx recombinant plasmid construct

The DNA sequence encoding POx from *Phanerochaete chrysosporium* (GenBank: aAS93628.1) was optimized for *Escherichia coli* expression using the online tool OPTIMIZER (*Puigbò et al., 2007*). The optimized sequence was synthesized and cloned into a pET-21b(+) vector between NdeI and XhoI restriction sites at GenScript (Piscataway, NJ).

### Expression and purification of POx

The recombinant pET-21b(+) plasmid with a confirmed POx insert sequence was used to transform competent BL21(DE3) *E. coli* cells (Life Technologies). A freshly transformed *E. coli* colony was used to inoculate 5 ml of LB medium supplied with 100 μg/ml ampicillin. The culture was grown in an incubator shaker at 37 °C for 4 h at 250 rpm, then it was transferred into 500 ml of the same growth medium and shaken at 37 °C and 220 rpm. Once the optical density of the culture reached $OD_{600} = 0.5$, the temperature was lowered to 25 °C, and after 1 hour protein expression was induced by the addition of 0.1 mM isopropyl β-D-1-thiogalactopyranoside (IPTG). The induced cells were incubated at 25 °C and 220 rpm overnight and harvested by centrifugation at 4,000 g for 20 min. Harvested cells were resuspended in 30 ml of 50 mM sodium phosphate buffer at pH 7.8 with 300 mM NaCl, 10 mM imidazole and 0.1 mM PMSF supplied with an EDTA-free complete Protease Inhibitor Cocktail tablet (Roche, Mannheim, Germany). The cells were disrupted by sonication on ice for a total of 10 min. Cell lysate was cleared of cell debris by centrifugation at 16,000 rpm (Beckman JA-17 rotor) for 30 min at 4 °C and mixed with 15 ml of Qiagen Ni-NTA agarose. The suspension was shaken at 4 °C for 1 h, loaded onto a column and washed with 6 bed volumes of 50 mM sodium phosphate buffer at pH 7.8 with 300 mM NaCl, 10 mM imidazole, 0.1 mM PMSF at 4 °C. The resin was washed (1) with 50 mM sodium phosphate buffer at pH 7.8 containing 300 mM NaCl and 50 mM imidazole, and (2) with 50 mM sodium phosphate buffer at pH 7.8 containing 300 mM NaCl and 100 mM imidazole. The POx protein was eluted with 50 mM sodium phosphate buffer at pH 7.8 with 300 mM NaCl and 250 mM imidazole and was kept either in ice or flash-frozen in liquid nitrogen with 10% glycerol and stored at −80 °C until use. For further experimentations, stock POx was dialyzed against 50 mM sodium phosphate buffer at pH 7 with 500 mM NaCl.

Protein concentration was determined using bicinchoninic acid assay (Thermo Scientific, Waltham, MA) according to the manufacturer's protocol. Molecular mass calculations were done using ExPASy ProtParam tool (*Gasteiger et al., 2003*). The calculated molecular mass of full-length POx was 72,198 Da and the molecular mass of proteolyzed POx was 66,529 Da, considering that fragments ii, iv and vi (Fig. 1D) remain intact after limited proteolysis.

### Molecular Dynamics Simulations (MDS)

The crystal structure of POx (PDB ID: 4mig, chains A, B, C and D) was used for molecular dynamics simulations. Loop polypeptide fragments between peptide bonds that were assumed to be cleaved by trypsin were removed from the chain A *in silico,* and a pdb file for proteolyzed POx was generated using UCSF CHIMERA 1.9 (*Pettersen et al., 2004*).

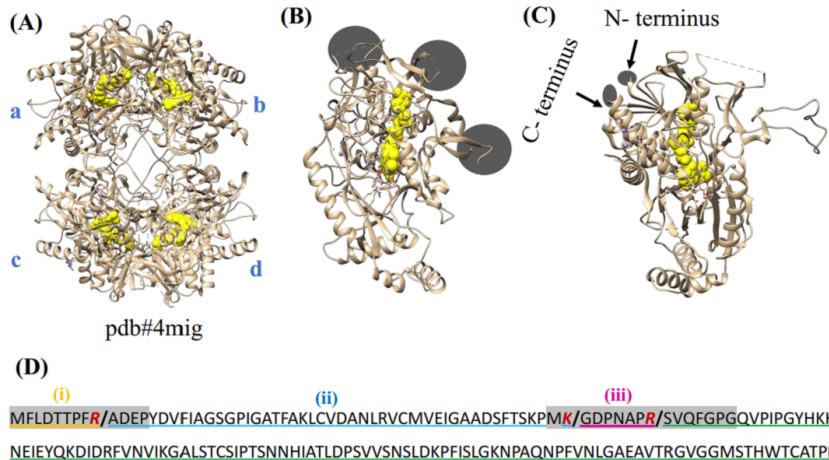

**Figure 1** **Crystal structure of POx and prediction of proteolytic fragments from trypsin treatment.** (A) POx homo-tetramer with four subunits labeled as A, B, C and D (pdb:4mig); (B, C) two projections of the ribbon model of wild-type POx monomer with FAD shown in yellow. Circles and arrows show the locations of the exposed regions and the terminal flexible regions, respectively; (D) POx sequence shows exposed flexible regions and N- and C- terminal regions highlighted as gray. The residues Arg (R) and Lys (K) representing potential cleavage sites are marked as red. Limited trypsinolysis is predicted to result in 7 fragments (separated by slash '/' at the cleavage sites). Three proteolytic fragments (ii- blue, iv-green and vi-red) expected to remain together as the core of POx, while other fragments (iii-magenta, v-black, i-orange and vii-purple) are expected to be removed.

Simulations of the proteolyzed POx tetramer were conducted using AMBER 11 (*Case et al., 2010*). Hydrogen atoms were first added and the peptide was then placed in a simulated box of TIP3P water molecules with a 10 Å minimal distance from the outermost side chains to the edge of the box. The charge of the protein was neutralized by adding $Na^+$ ions in the protein-water simulated box. The system was energy-minimized to overcome the effects of steric overlap between atoms. The motion of each peptide was simulated as a function of time using the SHAKE algorithm (*Ryckaert, Ciccotti & Berendsen, 1977*) with a time step of 2 femtoseconds. The simulations were run at 300 K for 80 nanoseconds in order to reach steady states. Structural comparison was done using UCSF CHIMERA 1.9.

## Limited proteolysis

To remove the flexible unstructured regions of POx structure, limited proteolysis was performed using trypsin. Trypsin is a highly specific serine protease, which cleaves peptide chains at the carbonyl side of lysine and arginine. The limited proteolysis was done in 50 mM sodium phosphate buffer at pH 7, containing 500 mM NaCl, at room temperature. POx to trypsin mass ratio of 50:1 was used. At several time points, 10 µl aliquots were removed, mixed with the sample buffer (2x Laemmli buffer composition: 0.125M Tris-HCl,

20% Glycerol, 4% SDS, 0.004% bromophenol blue, 10% 2-mercaptoethanol) for analysis and boiled to terminate trypsinolysis. The results were analyzed by sodium dodecyl sulfate polyacrylamide gel electrophoresis (SDS-PAGE). Molecular masses of protein bands were calculated using Image Lab software (BioRad).

## Purification of POx proteolytic fragments

To prepare large quantities of proteolytic fragments for further enzyme testing, trypsinolysis was terminated by adding 1 mM Pefabloc in 5.5 h. Proteolytic fragments were purified either by Fast Protein Liquid Chromatography (FPLC) using a QFF anion exchange column (for CV experiments) or Size Exclusion Chromatography using a PD-10 desalting column (GE Healthcare Life Sciences, Pittsburgh, PA) (for cross-linking and CD experiments). For FPLC based purification, we used 20 mM Tris and 1mM DTT at pH 8.5 with 0–1 M NaCl gradient to elute samples. POx fragments were eluted from 0.30 M to 0.36 M NaCl concentration range. For desalting column based purification, we used 50 mM sodium phosphate buffer at pH 7, containing 500 mM NaCl.

## POx cross-linking with glutaraldehyde

For cross-linking experiments, we used full-length POx and proteolyzed POx (0.415 mg/ml) in 50 mM sodium phosphate buffer at pH 7, containing 500 mM NaCl. Full-length POx was treated with 0.02% (v/v) glutaraldehyde at room temperature and 10 µl aliquots were removed at 10 and 60 min time points. Crosslinking reaction was stopped by adding 10 µl of the sample buffer for SDS-PAGE. POx fragments were treated with glutaraldehyde (GA) (0.05 % and 0.10 % (v/v)) at room temperature and 10 µl aliquots were removed at 30 and 60 min time points. Crosslinking reactions were stopped by adding 10 µl sample buffer for SDS/PAGE.

## Size exclusion chromatography

For size exclusion chromatography of POx, we used Superdex 75 10/300 GL column (GE Healthcare, Pittsburgh, PA). 200 µl (∼0.4 mg/ml) of a sample was loaded for each protein. For eluting the protein samples, we used 50 mM phosphate buffer at pH 7, containing 150 mM NaCl, at flow rate of 0.5 ml/min.

## Circular dichroism (CD) measurements

CD spectra of 0.1 mg/ml full length POx and proteolyzed POx in 12.5 mM sodium phosphate buffer at pH 7 with 125 mM NaCl were recorded at 20ºC from 195 to 250 nm at 0.5 nm intervals, on an Aviv CD spectrometer Model 400 (Aviv Biomedical Inc., Lakewood, NJ). Hellma Analytics (Plainview, NY) quartz cuvettes with one mm path length were used for CD measurements.

## Enzyme activity assay

Color reagent for activity assay was prepared by mixing 20.3 mg 4-aminoantipyrine, 9.5 mg Phenol, 1 mg Peroxidase, 10 ml Tris-HCl buffer (0.1 M, pH 7) in a light-protected tube. For enzyme activity assay, 800 µl deionized water, 500 µl Tris-HCl (0.1 M, pH 7), 100 µl color reagent, 50 µl of 1 M glucose, 50 µl 0.02 mg/ml POx sample were added and mixed thoroughly. During the reaction, quinoneimine dye was produced which absorbs

light at 500 nm. The rate of production of quinonimine dye is directly proportional to the enzymatic oxidation of glucose. The associated reactions associated in the experiment are:

$$\text{D-Glucose} + O_2 \rightarrow 2\text{-Dehydro} - D\text{-glucose} + H_2O_2$$

$$2H_2O_2 + 4 - \text{Aminoantipyrine} + \text{Phenol} \rightarrow \text{Quinoneimine dye} + 4H_2O.$$

The absorbance of the reaction mixture was recorded at 500 nm for 4 min. We performed 10 enzyme activity assay measurements for three separately obtained samples of proteolyzed POx and reported the mean and standard deviation as error bars in the corresponding figure.

### Statistical analysis

We ran two tailed statistical $t$-test between the two groups of enzyme activity results (10 experiments per replicates $\times 3$ replicates each $= 30$ data points for each) of wild-type and modified POx. The test result gave the $p$-value of 0.0079. We considered $p$-value below 0.05 as statistically significant difference.

## RESULTS

### In silico removal of exposed flexible regions in POx had no significant effect on POx structure

Initial inspection of the POx amino acid sequence and its three-dimensional structure (pdb# 4mig) demonstrated presence of several exposed disordered regions. Disordered flexible regions occupy more space than ordered ones, and we hypothesize that by removing them we will be able to reduce distance between the surface of the protein and its active site. We chose limited proteolysis by trypsin to remove exposed flexible regions of POx with the purpose of overall structural reduction. Trypsin cleaves peptide chains at the carbonyl side of lysine (K) and arginine (R). Exposed disordered and flexible regions in proteins are typically the most susceptible to proteolytic cleavage. Removal of the exposed flexible regions will cut the protein primary sequence into several fragments (Fig. 1D), but given the high stability of the intact POx, the secondary and tertiary structural components (e.g., alpha helices, beta sheets, hydrogen bonds, disulfide bonds) may still maintain the overall globular protein conformation.

To predict the effects of trypsinolysis on POx conformation, we analyzed the crystal structure of POx shown in Fig. 1. We identified five exposed disordered regions in the POx structure. The regions are Met1-Pro13 (N-terminal residue), Met55-Gly70, Leu307-Ser319, His345- Pro374, and Arg618-His629 (C-terminal residue), all highlighted in gray in Fig. 1D. As trypsin is highly specific in cleaving after Arg and Lys residues, we identified potential trypsin cleavage sites within these exposed disordered regions (Fig. 1D). Figure 1D also shows the sequences of POx tryptic fragments that we predicted to be produced after limited proteolysis. Upon treatment with trypsin, two exposed flexible fragments, residues Gly57-Arg63 and Gly361-Arg367 (iii-magenta and v-black, respectively), together with the N- (i-orange) and C- (vii-purple) terminal flexible fragments were expected to be removed from the POx structure. Although the removal of these four exposed flexible

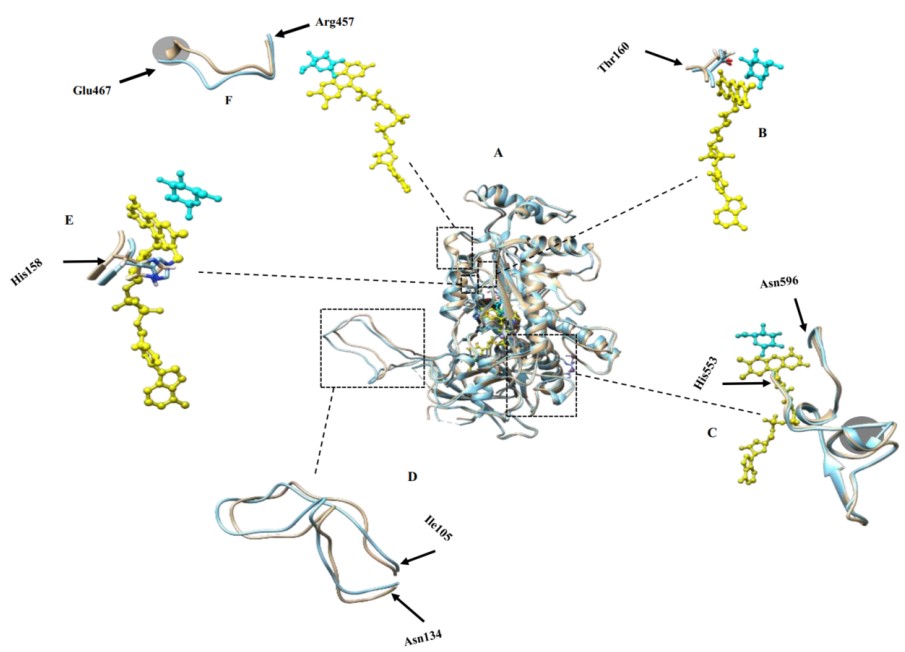

**Figure 2** **Molecular dynamic simulation results.** (A) Superimposed structures of monomers in simulated tetramers with (brown) and without modifications (blue); (B) Thr160 crucial for catalysis and binding to the co-factor; (C) active site residues comprising His553-Asn596 that participate in catalyzing the reaction of glucose oxidation; (D) oligomerization arm comprising residues Ile105-Asn134 that participate in POx tetramer formation, (E) His158 crucial for catalysis and binding to the co-factor; (F) dynamic substrate recognition loop comprising residues Arg457-Glu467 important for substrate recognition and binding. FAD is labeled yellow and substrate is labeled cyan. Gray circled regions in (C) and (F) show some structural differences of the modified POx compared to wild-type POx.

regions will create gaps within the POx structure, we expected the other three fragments (ii- blue, iv-green and vi-red) to keep most of the non-covalent contacts existing in the intact protein. Therefore, we assumed that these three fragments would remain bound together in a compact globular structure and maintain the surrounding of the central FAD molecule in a way appropriate for efficient enzymatic catalysis.

To mimic the trypsinolysis *in silico*, we removed the chosen regions (region i, iii, v, vii in Fig. 1D) from the wild-type POx. Then, we performed molecular dynamic simulation for the modified POx tetramer molecules to predict the effect of trypsinolysis on the modified POx conformation. The final simulated structure of the modified POx showed very small overall backbone shifts with respect to the wild-type POx (Fig. 2A). These shifts had a root mean square deviation (rmsd) over C-α atoms of 1.26 Å, which is comparable to the rmsd (1.07) calculated for the wild-type POx with respect to a homologue of POx from different organism *Trametes multicolor*. Therefore, we did not expect to significantly alter the structure of the POx active center and thus the functionality of POx.

We analyzed the positions of the residues that are important for the catalysis of glucose oxidation. The residues from His553 to Asn596 (Fig. 2C) are active site residues and residues from Arg457 to Glu467 (Fig. 2F) are dynamic substrate recognition loops that are important for the catalysis and substrate recognition (*Hassan et al., 2013*). We compared these two
important regions in the wild-type and modified POx structures and found no significant differences. There were slight backbone shifts with rmsd over C-α atoms of 0.754 Å and 2.48 Å respectively, and helices formation in regions Thr564-Lys566 and Val465-Glu467 (shown in gray circles in Figs. 2C and 2F respectively, rmsd calculated between C-α atoms for these circled regions are 0.633 Å and 3.45 Å respectively). All these differences are far away from the FAD (yellow) and substrate binding sites (cyan). For this reason, we expected that these changes will not have significant effect on enzyme functionality. In addition to that, His158 and Thr160 are reported important in catalysis and binding to the FAD group (*Salaheddin et al., 2010*). Figure 2E shows His158 residue and Fig. 2B shows the Thr160 residue where wild-type (blue) and modified (brown) proteins are superimposed to compare their side chain conformations. There is no significant difference other than small backbone shift in His158 (rmsd over C-α atoms = 0.827 Å) and Thr160 (rmsd over C-α atoms = 1.133 Å) residues.

We also compared positions of the POx residues that are reported to participate in tetramer formation. A long oligomerization arm comprising residues Ile105 to Asn134 helps to form POx homotetramer (*Hassan et al., 2013*). Figure 2D shows superimposed oligomerization arms of wild-type (blue) and modified (brown) POx structure. We observed a small backbone shift with the rmsd of 1.82 Å between C-α atoms that may not be significant enough to alter the tetramer formation. These MDS results suggested that removal of disordered regions should not significantly affect the active center or formation of a tetramer and therefore should not affect the enzyme functionality.

## Limited proteolysis of POx resulted in formation of several fragments

Based on our prediction, limited proteolysis by trypsin will result in formation of seven fragments with theoretical molecular masses of fragment (i) through (vii): 1.127, 4.907, 0.725, 33.154, 0.618, 28.503, 1.589 kDa, respectively (Fig. 1D), three of which (fragments ii, iv and vi) will maintain a globular polypeptide structure, whereas the rest (fragments i, iii, v and vii) will be separated and exist as unbound and unstructured polypeptides in solution.

To find conditions for removal of flexible regions, trypsin was added to POx at 1:50 w/w ratio and proteolysis was stopped in 3, 10, 30 and 60 min using protease inhibitor pefabloc. These are the typical times used for limited proteolysis of proteins (*Kostyukova et al., 2000*). Within the first three minutes , a band of ∼65 kDa appeared corresponding to a ∼5 kDa decrease in the POx molecular mass (Fig. 3A, lane 3). By the 60-minute time point, further proteolysis of 65 kDa fragment resulted in the formation of two major bands with molecular masses of 34 kDa and 27 kDa and a minor band of 32 kDa (Fig. 3A, lane 6). At this point we performed an enzyme functionality assay and found that the enzyme was still highly active.

Next, we decided to increase the time of proteolysis until the 65 kDa band disappears. For that, limited trypsinolysis was repeated and stopped at 4, 5, 5.5, 6, 6.5 and 7 hours. After 5 hours of treatment, most of the 65 kDa band disappeared and the bands with the molecular masses of 32 and 27 kDa became the major ones, while the band with the MW of 34 kDa became a minor band. This mixture of proteolytic fragments was very stable

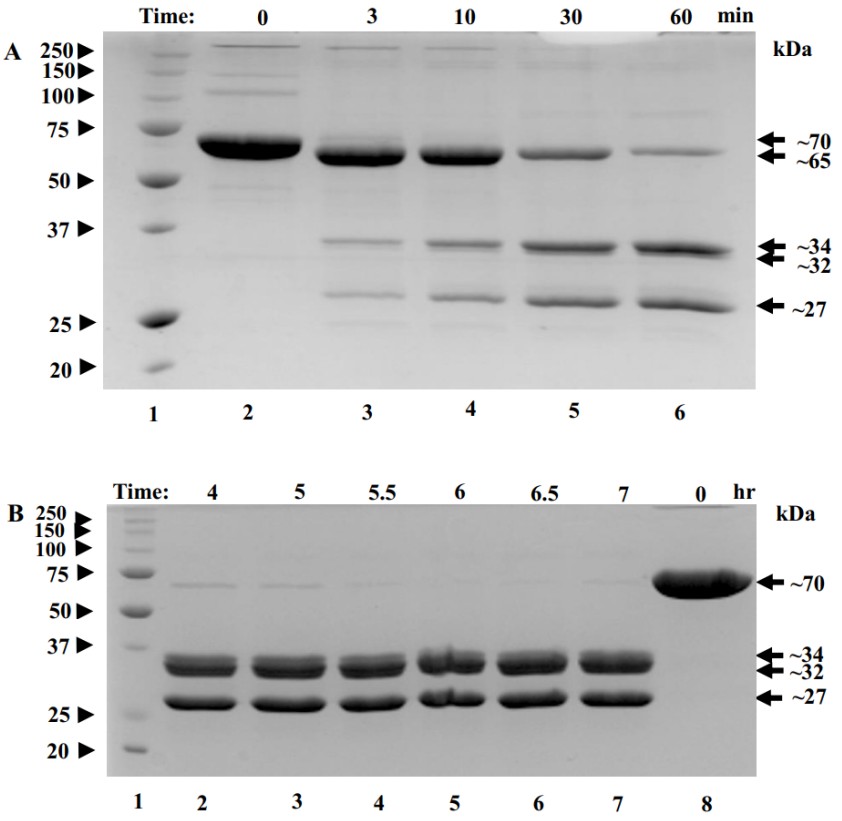

**Figure 3** SDS-polyacrylamide gel (12% acrylamide) shows time-course of the limited proteolysis of POx by trypsin (50:1 w/w). (A) Molecular mass standards (lane 1); full-length POx (lane 2); POx treated with trypsin for different time points: 3 min (lane 3), 10 min (lane 4), 30 min (lane 5), 60 min (lane 6); (B) Molecular mass standards (lane 1); POx treated with trypsin at different time points: 4 hr (lane 2), 5 h (lane 3), 5.5 h (lane 4), 6 h (lane 5), 6.5 h (lane 6), 7 hr (lane 7); full-length POx (lane 8). Arrowheads indicate standard proteins, arrows indicate POx (full-length and fragments).

and did not undergo any further digestion for up to 7 hours (Fig. 3B). Also, the reaction mixture was tested for POx functionality and it was found highly active.

## Theoretically predicted molecular masses of tryptic fragments matched with actual molecular masses

Comparing molecular masses of the obtained fragments with the predicted molecular weights (Fig. 1D) and considering a ∼5% uncertainty in determining molecular mass of a protein using SDS-PAGE, we interpreted the 5 kDa decrease in molecular mass appearing in the early stages of the trypsin digestion as originating from the removal of the N- and C- terminal flexible regions of the protein (fragments i and vii). The predicted combined molecular mass of the N- and C- terminal regions is 2.716 kDa, which is less than the observed 5 kDa mass decrease. However, our interpretation is consistent with the fact that the removal of the terminal regions should result in a single polypeptide and therefore a single band as in Fig. 3A (lane 3).

The 34 kDa fragment corresponds to the predicted tryptic fragment-iv (green) flanked by the flexible region-iii (magenta) and the flexible region-v (black) (predicted combined molecular mass is 34.5 kDa). The 32 kDa band corresponds to the predicted tryptic fragment-iv (green) (predicted molecular mass 33.2 kDa) after the flexible regions iii and v were removed from the 34 kDa fragment. The 27 kDa major band corresponds to the structured region vi (red) (theoretical molecular mass 28.5 kDa) (Fig. 1D). Fragments with molecular masses smaller than 5 kDa typically cannot be seen in 12 % SDS-polyacrylamide gels due to their high mobility, fast diffusion and poor staining. Therefore, we did not expect to observe fragments that correspond to structured region ii, and flexible regions i, iii, v, vii.

## POx tryptic fragments form a tetramer

Our MDS data indicated that removing the flexible regions by limited proteolysis should not affect the tetramer formation. Also during limited proteolysis, the oligomerization arm should not be affected by trypsin, since this segment is not exposed to the outer surface of POx. To verify experimentally that the tryptic fragments still form a tetramer, we cross-linked the protein by glutaraldehyde. Cross-linking by glutaraldehyde is often used to obtain preliminary information on quaternary structure of a protein (*Fadouloglou, Kokkinidis & Glykos, 2008*). When protein oligomers are treated with glutaraldehyde cross-linker, the polypeptide chains form inter-subunit covalent cross-links holding the chains together in a denaturing environment. These cross-linked chains can be further examined by SDS-PAGE and polypeptide chains interacting with each other can be identified. To perform this experiment, both full-length POx and POx fragments after 5.5 hours of trypsinolysis (denoted as proteolyzed POx thereafter) were crosslinked with different amounts of glutaraldehyde (0.02%, 0.05%, 0.1% v/v) and analyzed by SDS-PAGE.

SDS-PAGE image in Fig. 4 shows that, upon glutaraldehyde treatment, the proteolysed POx and the uncleaved full-length POx formed bands with molecular masses higher than a single polypeptide chain of the uncleaved POx. For the uncleaved POx, we observed bands with the molecular mass of 70, 100, 126, 220 and a band above 250 kDa (Fig. 4A). Based on the sequence of POx predicting the molecular mass of a POx monomer as 70 kDa, the band >250 kDa corresponds to a cross-linked POx tetramer. Bands with lower molecular masses could represent cross-linked dimers and trimers. For the POx cleaved with trypsin, we observed bands with the highest molecular mass of above 250 kDa. Similar to the full-length POx, we assigned the band >250 kDa in Fig. 4B to a tetramer. The bands with lower molecular masses could correspond to intermediate cross-linked complexes of the proteolytic fragments.

We also performed size exclusion chromatography to estimate and compare the molecular sizes of the full-length and proteolyzed POx. Our results (Fig. 5) showed that the two POx elution profiles overlapped with each other confirming that they have similar molecular sizes that were not resolved on the column. The data we obtained from the glutaraldehyde treatment and the chromatography, therefore, show that after proteolysis the inter-subunit interactions in POx were not affected and the protein exists in solution in a tetrameric form.

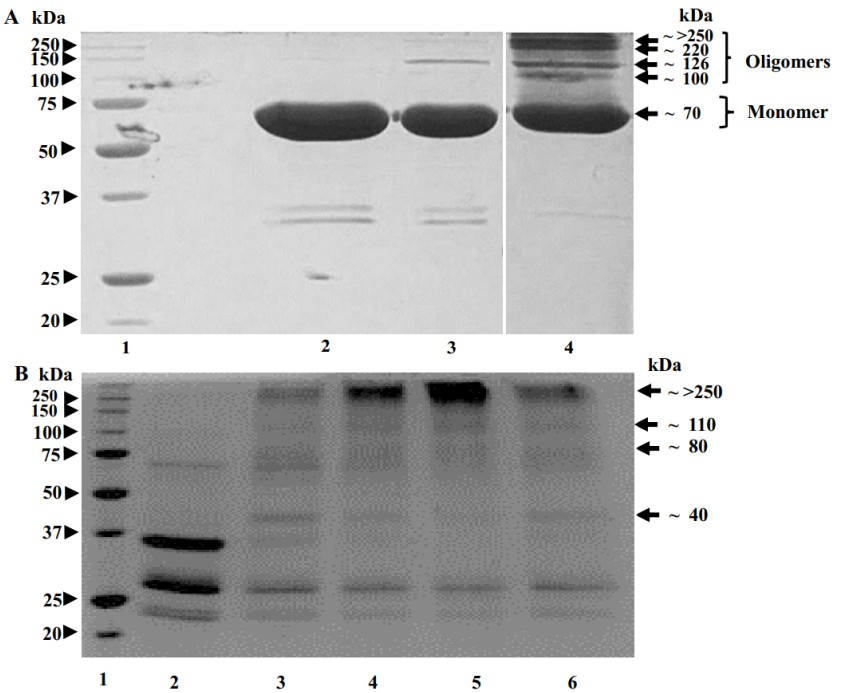

**Figure 4** **Results of crosslinking of POx by glutaraldehyde (GA) shown by SDS-polyacrylamide gel electrophoresis.** (A) Time dependence of crosslinking full-length POx by 0.02% GA: molecular mass standards (lane 1); full-length POx (lane 2), full-length POx treated for 10 min (lane 3) or for 60 min (lane 4). (B) Time dependence of crosslinking POx after 5.5 h of trypsinolysis by 0.05 or 0.1% of GA: molecular mass standards (lane 1); proteolyzed POx (lane 2) proteolyzed POx treated for 30 min with 0.05% (lane 3) or 0.1% GA (lane 4), and for 60 min with 0.1% (lane 5) or 0.05% GA (lane 6). Arrowheads indicate molecular mass standard proteins, arrows indicate cross-linked polypeptides.

Elution profiles overlap for full-length and proteolyzed POx.

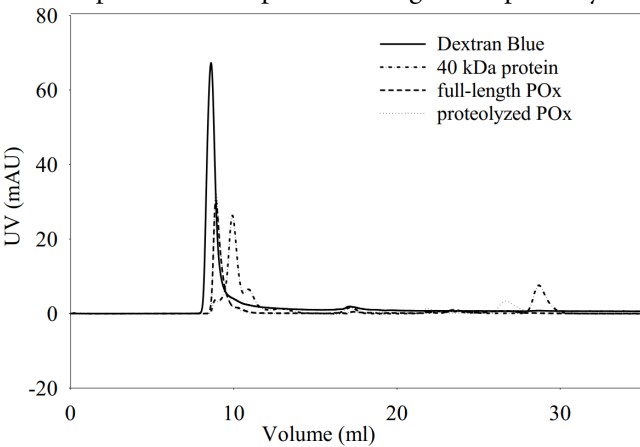

**Figure 5** **Elution profiles for dextran blue, full-length POx, proteolyzed POx after 5.5 h of trypsinolysis, and tropomodulin (40 kDa) obtained using size exclusion chromatography.**

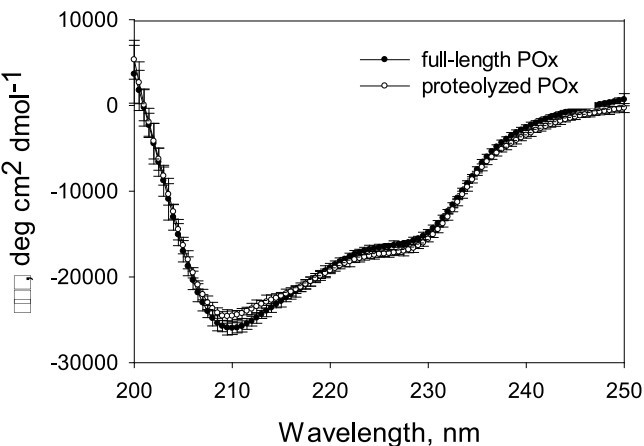

**Figure 6**  CD spectra measured for full-length and proteolyzed POx (after 5.5 hours of trypsinolysis) in 12.5 mM sodium phosphate buffer, pH 7, 125 mM NaCl.

## Circular dichroism (CD) spectra analysis showed no increase in disorder after proteolysis

Based on our MDS results, we expected that the proteolysis does not damage the enzyme structural conformation and therefore its functionality. Gel filtration data and the SDS-PAGE analysis of the proteolytic reaction mixture and the products of the glutaraldehyde treatment suggested that the limited trypsinolysis removed disordered regions of POx, while the globular structure of the POx core remained unaffected. Therefore, we expected that there would be no increase in the disordered secondary structure content of proteolyzed POx. CD spectroscopy is a powerful technique to study changes in the secondary structure content of any proteins. To prepare a sample of proteolyzed POx for CD analysis, we separated small cleaved fragments from the globular POx core by size exclusion chromatography. Figure 6 shows the CD spectra of the full-length and purified proteolyzed POx. There are no drastic changes in the spectrum, however, intersection of the spectrum of the proteolyzed POx with the *x*-axis slightly shifted to higher wavelength compared to the spectrum of wild-type POx. This indicates the slight increase in ordered secondary structure content and suggests that the main core of cleaved POx maintained its native three-dimensional structure after removal of disordered regions during limited trypsinolysis.

## Proteolyzed POx is functionally active

Enzymatic activity is highly dependent on the structural conformation of its active site. Using CD, size exclusion chromatography and cross-linking we demonstrated that proteolysis did not affect the secondary and the quaternary structures of the enzyme, and therefore, most probably the tertiary structure, including the active site conformation, remains unchanged. In such case, the proteolyzed POx should still be functional. To test if removal of disordered flexible regions has an impact on its enzymatic activity, we measured the POx activity after proteolysis and compared it with the wild-type POx activity. The rate of enzymatic reaction was measured using a coupled reaction method that

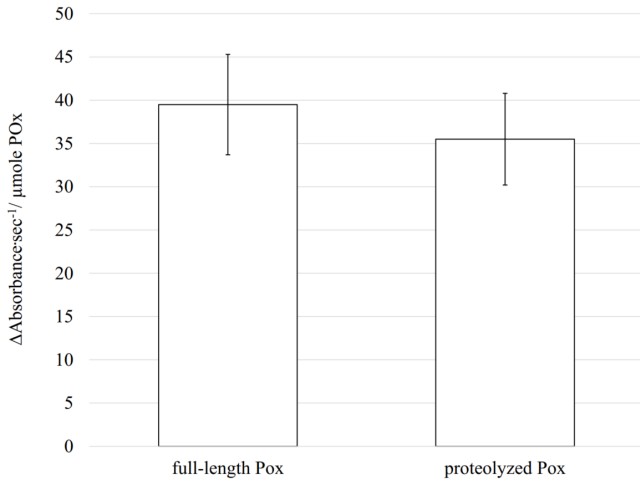

**Figure 7 Enzymatic activity measured for full-length and proteolyzed POx (after 5.5 h of trypsinolysis) shows that about 90% of the activity is retained after proteolysis.** The error bars represent the standard deviation ($n = 30$).

produces quinoneimine dye. Oxidation of glucose is directly coupled to the quinonimine dye production and the rate of change in quinoneimine dye concentration was a direct measure of the enzymatic activity. The activity test showed that the wild-type POx had specific activity of $39.5 \pm 5.8 \, \Delta$ Absorbance s$^{-1}$/ μ mole POx whereas modified POx had specific activity of $35.5 \pm 5.3 \, \Delta$ Absorbance s$^{-1}$/ μmole POx (the absorbance of the quinoneimine dye was measured at 500 nm). Most of the enzyme activity (∼90 %) was retained after trypsin treatment (Fig. 7) confirming that there are no drastic changes in the active site conformation.

### Ethics

The study was carried out in accordance with the policies of the WSU Institutional Biosafety Committee (approval reference number 01131).

## DISCUSSION

One of the recent studies has performed deglycosylation of an enzyme named glucose oxidase (GOx) to reduce the hydrodynamic diameter of the enzyme molecule and therefore the critical separation distance between active site and the surface (*Courjean, Gao & Mano, 2009*). As a result, the GOx active site was brought closer to its surface, and the electron transfer efficiency was significantly improved. According to the Marcus theory (*Marcus & Sutin, 1985*), the kinetics of direct electron transfer between the active site of an enzyme and the electrode surface is highly dependent on their separation distance. The probability for the electrons to "jump" from the active site of the enzyme to the electrode increases exponentially as the separation distance decreases. Therefore, for the proteins with deeply buried active sites, an approach involving protein size minimization can be a great solution to improve their electron transfer efficiency.
The difficulty with protein size minimization lies with the fact that enzymes are highly structurally organized molecules. Their three-dimensional (tertiary) globular structure is defined by their amino acid sequence, or primary structure, which evolved over considerable time. When enzyme primary structures are modified from their natural form, some important intra-molecular interactions can be lost, which can change the overall conformation (tertiary structure) or even lead to a complete loss of structure rendering them unfolded. This will lead to the decrease or complete loss of their enzymatic activities. Consequently, the protein engineering effort aiming at the protein size minimization requires taking into consideration major forces holding the protein in its globular folded state and keeping the active site operational.

The removal of the flexible disordered regions of POx is an initial effort of the long-term goal of an enzyme structure minimization process to improve the electron transfer efficiency. In this study, we identified several exposed disordered regions in the POx structure. Our molecular dynamic simulation results suggested that removal of these disordered regions from the wild-type POx should not significantly affect its active center or formation of its tetramer structure and therefore should not affect its functionality. The limited proteolysis of the POx was then performed using trypsin to remove the identified flexible structural regions. As our molecular dynamic simulation predicts, the experimental data obtained from the glutaraldehyde treatment and the chromatography show that after proteolysis the inter-subunit interactions in POx were not affected and the protein exists in solution in a tetrameric form. Circular dichroism spectra analysis also indicates that there is no decrease in ordered secondary structure content and confirms that the main core of cleaved POx maintained its native three-dimensional structure. The enzymatic activity of the modified POx showed only 10% reduced activity compared to the wild-type POx. Hence, we demonstrated that limited proteolysis removed the disordered regions from the POx structure, while the protein core remained structurally intact, stable and catalytically active. Enzyme structure minimization may be done not only using limited proteolysis but also by genetic engineering to remove secondary structure elements that may not affect active site formation. In this study, we removed not entire loops/termini but 34 residues only (5.4% of total protein mass) due to trypsin specificity. In future studies, loops can be removed by changing them to short linkers (2–3 residue) by modifying the nucleotide sequence. This means that up to 70 residues can be removed and replaced by 6–9 residues resulting in 10% decrease of the enzyme molecular mass. In addition, flexible loops occupy more space then ordered elements and the decrease in size may be even more than 10%. MDS is a great tool to test results of such modifications. Modifications may decrease enzymatic activity, therefore the big challenge will be to ensure that any loss of enzymatic activity resulting from protein minimization is much smaller than the increase in the electron transfer efficiency.

# ACKNOWLEDGEMENTS

We thank Natalia Moroz for helping in POx expression and Kyle Swain for helping in POx activity test.

### Funding
This study was supported by Grand Challenges Seed Grant from Office of Research, WSU, to SH and ASK. The funders had no role in study design, data collection and analysis, decision to publish, or preparation of the manuscript.

### Grant Disclosures
The following grant information was disclosed by the authors:
Grand Challenges Seed Grant from Office of Research, WSU.

### Competing Interests
Alla S. Kostyukova is an Academic Editor for PeerJ.

### Author Contributions
- Tanzila Islam performed the experiments, analyzed the data, prepared figures and/or tables, authored or reviewed drafts of the paper, and approved the final draft.
- Catherine Booker and Dmitri Tolkatchev performed the experiments, authored or reviewed drafts of the paper, and approved the final draft.
- Su Ha and Alla S. Kostyukova conceived and designed the experiments, analyzed the data, authored or reviewed drafts of the paper, and approved the final draft.

### Ethics
The following information was supplied relating to ethical approvals (i.e., approving body and any reference numbers):

Washington State University granted Biosafety approval to carry out this study within its facilities (1131).

### Data Availability
The raw measurements are available in the Supplementary Files.

### Supplemental Information
Supplemental information for this article can be found online at http://dx.doi.org/10.7717/peerj-matsci.7#supplemental-information.

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
