# Peer review of "Limited proteolysis of pyranose 2-oxidase results in a stable and active complex"

_PeerJ Materials Science, doi:10.7717/peerj-matsci.7_

## Round 0.1 · original submission · Major Revisions

Please attend to the queries of the reviewers in a detailed rebuttal letter and revision.

Reviewer 1 ·

Basic reporting

The manuscript presented in a clear form, well referenced and good context. The work flow is straightforward and results are well described.

Minor comments:
Line 123: 53 check citation format

Figure 3 has an embedded legend as "Figure 1", a possible editing error.

Figure 3 a) and b) are mainly the same at different time points. Both can be a single figure, maybe merging lines 2 and 7 from 3 b).

Legend of Figure 4 is confusing and should be rewritten for clarity.

Figure 5 will be clearer if full length and proteolyzed POx lines are shown with solid lines, whereas markers in dashed or thinner lines.

Experimental design

. Did secondary structure content analysis of full and proteolyzed POx match with other results? Authors discussed secondary structure content, but values are not presented nor the deconvolution method used is described. How do the HT values behave? How many spectra were recorded or averaged? CD measurements are valuable for testing the main objective of this research, but authors use a visual inspection of the CD spectra as evidence.

Did the authors made blank reactions on enzyme activity without POx? I think an enzyme kinetics approach will give an insight of the effect of removing loops to the enzyme's turnover or substrate affinity.

Validity of the findings

The major goal of this research is to reduce the enzyme's size, without losing its enzymatic activity for further applications. But the authors do not expect that proteolysis by trypsin alters the active site or its surroundings significantly. Actually, they assume that two small loops of six residues are removed. I think it is not clear whether this is a proof of concept if the protein is not significantly size reduced or its active site is more exposed. In fact, the main finding is that two flexible/exposed loops can be removed without singnificantly loss of POx function.

Additional comments

From the biochemical approach of the experimental work, this manuscript is more suited for the biological and environmental section of this journal, than the materials science one.

Reviewer 2 ·

Basic reporting

Islam et al present the effect of reducing the size of pyranose 2-oxidase on stability and enzymatic activity. Using trypsin to remove flexible structural regions; however, most of the claims about this being a superior method and tool for electrochemical applications do not seem to be adequately supported. Lastly, the presentation of the experimental data needs to be substantially improved.

Experimental design

- The manuscript needs information on replicates and analysis. For all experiments, please provide number of replicates and what the errors are. This is critical for evaluation of the data. This needs to be fully evaluated and the data shown.

- The authors present some potentially interesting data in this work, however, there is no detailed biochemical and biophysical analysis of the proteolyzed enzymes. Some data are not clear with awkward phrases here and there.

- Only 1 proteolyzed enzyme was showed similar kinetic rate with WT? The authors state that this method is effective. I would challenge this concept by the authors expressed and purified all proteolyzed enzymes, which is by no means a low-effort endeavor.

- Figure 7: I would like to see the kcat and KM values and the error in these values.

- Which step in the catalytic cycle of proteolyzed enzyme is rate-limiting? For completeness it would be good to present the full kinetics parameters of proteolyzed enzyme.

Validity of the findings

- This manuscript could benefit greatly by a more focused effort on biochemistry and biophysical properties of particular proteolyzed enzymes, or a broadening of the experimental effort by identifying proteolyzed enzymes with a clearer effect on performance.

- Author should explain why proteolyzed enzymes can convert similar efficiently with WT?

- Results are a bit overselling and there is relatively little discussion about the usefulness/success rate of the applied method.

- The paper attempting to rationalise the observations is rather speculative and it is not clear what we have learned from the current study about making small proteins stable.

---

## Round 0.2 · accepted · Accept

The manuscript has improved over the review rounds and it is now accepted at PeerJ.

Reviewer 1 ·

Basic reporting

No comment.

Experimental design

No comment.

Validity of the findings

No comment.

Additional comments

The authors take into account the reviewers comment. Their revised manuscript improved its clarity to state the major findings and future perspectives.

Reviewer 2 ·

Basic reporting

This revised version is more clearer finding than the one before.

Experimental design

-

Validity of the findings

-